# Predicting the risk of asthma development in youth using machine learning models

**Matthew Xie**[1,2], **Chenliang Xu**[3*]

**1** Pittsford Sutherland High School, Pittsford, New York, United States of America, **2** Pratt School of Engineering, Duke University, Durham, North Carolina, United States of America, **3** Department of Computer Science, University of Rochester, Rochester, New York, United States of America

\* chenliang.xu@rochester.edu

## Abstract

Asthma is a chronic respiratory disease characterized by wheezing and difficulty breathing, which disproportionally affects 4.7 million children in the U.S. Currently, there is a lack of asthma predictive models for youth with good performance. This study aims to build machine learning models to better predict asthma development in youth using easily accessible national survey data. We analyzed cross-sectional combined 2021 and 2022 National Health Interview Survey (NHIS) data from 9,716 youth subjects with their corresponding parent information. We built several machine learning models with various sampling techniques (under- or over-sampling) for asthma prediction in youth, including XGBoost, Neural Networks, Random Forest, Support Vector Machine (SVM), and Logistic Regression. These models were further validated using the 2023 NHIS data. We examined the associations of potential risk factors identified from both Random Forest and Least Absolute Shrinkage and Selection Operator (LASSO) with asthma in youth. Between the different sampling techniques, undersampling the major class (subjects without asthma) yielded the best results in terms of the area under the curve (AUC) and F1 scores for the different predictive models. The Logistic Regression performed the best with the undersampled data, yielding an AUC score of 0.7654 and an F1 score of 0.3452. Beside of some well-known risk factors for asthma development, such as gender and socioeconomic status, we have identified additional potential factors associated with asthma development in youth such as "took prescription medication in past 12 months", "age" and "general health status" which had the highest magnitude mean Shapley Additive exPlanations (SHAP) values of 0.094, 0.076 and 0.042. This study successfully built machine learning models to predict asthma development in youth with good model performance. This will be important for early screening and detection of asthma in youth.

**Data availability statement:** The datasets analyzed for this study are openly accessible from the National Health Institute Survey website: https://www.cdc.gov/nchs/nhis/data-questionnaires-documentation.htm.

**Funding:** The author(s) received no specific funding for this work.

**Competing interests:** The authors have declared that no competing interests exist.

## Introduction

Asthma is a chronic lung disease portrayed by deadly asthma attacks that result in wheezing, breathlessness, and chest tightness [1]. Compared to those without asthma, children and adolescents with asthma had additional $3,362.56 in annual medical expenditures [2]. Currently, the only protective measures to combat asthma and asthma attacks are medications and avoidance of triggers [3,4]. The American Lung Association identified several factors that may cause asthma, such as family history, occupational exposure, and smoking [5]. Furthermore, studies have shown that 80% of asthma cases arise during the first six years of a person's life [6]. Due to the lack of a cure, it is crucial that asthma development in youth can be predicted and detected early so that prevention and early intervention may take place.

Recent studies on asthma epidemiology have identified some causes of asthma that align with those identified by the American Lung Association and observed the differences in asthma prevalence in different age groups [6,7]. One study on the epidemiology of asthma in children and adults found that asthma is more prevalent in boys among children while among adults asthma is more prevalent in women, which is triggered by similar causes, namely, family history, occupational exposures, and environmental triggers [7]. Another study revealed that childhood asthma has more common remission compared to adults, and asthma mortality rates are higher in adolescents and young adults compared to young children and older adults [6]. The higher asthma mortality rate among adolescents underscores the need to address asthma with preventative measures.

The recent development of artificial intelligence techniques (especially machine learning and deep learning models), as well as the data availability of detailed patient information, empowers the prediction and risk assessment of various chronic diseases. For example, machine learning has been employed in predicting cardiovascular diseases [8–10]. The use of machine learning and artificial intelligence in asthma diagnosis has been touched upon and strongly suggested to improve clinical practices, especially in low-income areas [11]. Numerous studies utilized clinical data to diagnose asthma patients using simple machine learning models such as support vector machines, random forests, decision trees, and logistic regression [12–14]. These studies only had access to datasets with hundreds of patients while few had access to datasets with more than 1,000 patients [12,15]. While many machine learning models have been successfully built for predicting asthma exacerbations among asthmatic patients [16–24], fewer studies have been conducted on predicting asthma risk using various datasets especially publicly available data, such as the 2019 Michigan BRFSS data and a small dataset of 202 children from Ibn Sina Hospital Center in Morocco [8,9]. In these studies, the dataset was either too small, limited to a specific region, or relied heavily on clinical factors, which may limit model performance and implementation [9]. This is evident by the relatively poor model performance in the 2019 Michigan BRFFS data with the highest AUC score of 0.63 for the logistic regression. These predictive models' performance could be improved by using a larger dataset and employing more advanced data processing and machine learning techniques.

This study aims to build predictive models for asthma development in youth using the combined National Health Interview Survey (NHIS) 2021 and 2022 survey data, which ensures a large sample size. Furthermore, we employed different sampling techniques, including oversampling, undersampling, and both, to address the class imbalance issue (most survey subjects had no asthma). Two complementary methods, Least Absolute Shrinkage and Selection Operator (LASSO) and random forest models, were used to identify factors associated with asthma development in youth. Finally, different machine learning models, such as logistic regression and neural networks, were used for predicting asthma development in youth, and their model performances were compared.

## Materials and methods

### Data source

The National Health Interview Survey (NHIS) is a data collection program for the National Center for Health Statistics (NCHS), a part of the Centers for Disease Control and Prevention (CDC). The NHIS is a household interview survey that involves face-to-face interviews followed by telephone interviews throughout the year. The NHIS collects data about the health of the noninstitutionalized civilian population of all ages in the United States. The purpose of the NHIS is to monitor the health of the United States population by categorizing health trends by demographic and socioeconomic circumstances. The NHIS 2021, 2022 and 2023 children and adults data are publicly available from the CDC website [10].

### Data pre-processing

The outcome variable of this study is asthma, which is based on the question, "Has a doctor or other health professional EVER told you that you had asthma?" If the answer is "Yes," the subject is considered to have asthma. If the answer is "No," the subject is regarded as having no asthma.

As shown in S1 Fig, we merged the youth survey data with their linked parent data by matching the unique child's household ID with their respective parents, which was conducted for both 2021 and 2022 NHIS data, respectively. The merged 2021 NHIS dataset has 7,070 youth subjects, while the 2022 NHIS dataset has 6,261 youth subjects with their corresponding parent information. We combined the 2021 and 2022 NHIS data to increase the sample size to form the final dataset with 13,331 youth subjects.

To get the data ready for the machine learning models, we employed several preprocessing or cleaning steps (S2 Fig). According to the codebook for adult and youth NHIS data, we first selected relevant variables potentially related to asthma development according to previous literature [20–24] and excluded irrelevant variables in the subsequent analysis. For example, questions specific to the survey design or a specific population not related to asthma, such as cancer patients, were not included in the analysis. After the irrelevant variables were removed, we removed all variables with only one level (constant variables) or variables only focused on a subgroup of age (such as age 5–17) because this study focuses on youth aged 0–17. Once this step was completed, we deleted all records with missing values for the remaining variables and responses such as "don't know," "refused," or "not ascertained." The final dataset contains 9,716 youth subjects and 45 variables with their corresponding parent information, which we used for subsequent analysis. In addition, in order to compare the model performance for different age groups, we stratified our data into three age groups: 0–4, 5–12, and 13–17 years.

### Feature/variable selection

To identify important features for predicting asthma in youth, we employed two complementary machine learning models (LASSO and Random Forest) for feature selection (S2 Fig). LASSO selects important features by first establishing a penalized regression model where the dependent variable is equal to the sum of the independent variables multiplied by the estimated coefficients and the error term. LASSO essentially finds the coefficient values (while shrinking them toward

zero) that minimize the sum of the squared differences between predicted and actual values [25]. This process makes LASSO useful for removing irrelevant features. Random forest can be used for both classification and regression [26]. The random forest model evaluates the variables based on the aggregated outputs from individual decision tree models to select the most relevant features [26]. Each model generated a list of the most significant variables or features for asthma prediction. We used the mean of feature importance values for all variables as the cutoff to select the variables/features. To avoid potential multicollinearity problems, we examined the multicollinearity issues using the variance inflation factor (VIF) values. We removed the highly correlated variables with VIF values larger than 2. This resulted in 34 variables selected from LASSO, 13 of which were also chosen by Random Forest. To maximize the model performance, we utilized the union (34 variables) of two variable lists for machine learning models. However, to understand how these variables contribute to asthma prediction in youth, we took the intersection (13 variables) of the two lists to minimize the false positive.

### Class imbalance

Considering that the proportion of youth subjects with asthma in the dataset is relatively small, the class label for asthma in our dataset is unbalanced, with most subjects having no asthma, which can significantly influence the model performance of predictive models. To deal with the class imbalance, after randomly splitting the data into training and test datasets with a ratio of 7–3 (70% of the data for training and 30% of the data for testing), we applied different sampling techniques to balance the class labels in the training dataset. The sampling techniques include SMOTE-Tomek (both oversampling and undersampling), undersampling, and oversampling (SMOTE) [27]. The undersampling technique will randomly remove some samples from the major class (subjects without asthma) so that the number of subjects with asthma is similar to those without asthma. As an oversampling technique, SMOTE will create additional synthetic records for subjects with asthma so that the numbers of subjects with and without asthma are similar. SMOTE-Tomek is a sampling technique employing both oversampling and undersampling [28]. This combination of over and undersampling helps balance the cons of only performing under or oversampling.

### Building machine-learning models

In this study, we used multiple machine-learning models such as random forest, XGBoost, neural networks, support vector machine with a linear kernel, and logistic regression to predict asthma development in youth (S3 Fig). Our choice of models was primarily based on the predictive models used in similar studies, which include a variety of predictive models that utilized different prediction techniques [8,29,30].

**Random forest.** Random forest is a machine-learning modeling technique that can be used in both classification and regression [26]. A typical random forest model is composed of numerous decision trees, each of which classifies data points into different class labels depending on their features. Eventually, the random forest model takes the output of each decision tree model and chooses the best output. This model is good for large datasets such as the NHIS dataset and has high precision [31]. The optimal hyperparameters of our best random forest model are: 500 estimators, 3 minimum sample split, and 'entropy' criterion.

**XGBoost.** Similar to the tree-based random forest model, Extreme Gradient Boosting is a gradient-boosted decision tree [32]. However, instead of outputting the average of each decision tree output in the random forest, XGBoost outputs the weighted average of each decision tree output. XGBoost is characterized by a combination of weak decision tree models to create a strong combined model. An advantage of XGBoost is that it reduces bias and underfitting [32]. The optimal hyperparameters of our best XGBoost model are: 5 estimators and 'gbtree' booster.

**Feedforward neural networks.** A perceptron is a machine learning algorithm that has many nodes whose output is an input for another node [33]. Within each node, the neural network takes the sum of the product of the weight and value for each input, adds a bias value, and passes it over to an activation function. The resultant value is then passed to nodes in the next layer in the network. The high computational power and accuracy of feedforward neural networks make it

desirable for this study [34]. The optimal hyperparameters for our best neural networks for our dataset are: 64 units in the first hidden layer, 32 units in the second hidden layer, ReLU for hidden layers, Sigmoid for output layer, Adam optimizer, binary crossentropy for loss function, 10 epochs, 32 batch size, and.20 validation split.

**Support vector machine.** Support vector machine (SVM) is mainly used for classification but can also be used for regression analysis [35]. SVM involves the creation of a hyperplane or line that the model uses to distinguish between different groups of data points. After the SVM is run, the model outputs a number either greater than 1 or less than –1, with each scenario classifying the data point as a group. SVM is typically used for small datasets due to its high accuracy and long run times [36]. In this study, the linear kernel was used. The optimal hyperparameters for our best support vector machine for our dataset are: linear kernel, and 0.009181282042611968 for the 'C' value.

**Hyperparameters fine-tuning.** In this study, we used Optuna to implement Bayesian optimization to fine-tune the hyperparameters of the selected models [37]. For different predictive models, there are different hyperparameters that the user may adjust to improve the model's performance. For example, the hyperparameters for random forest include the number of trees, maximum levels within each decision tree, and the minimum number of datapoints allowed at each node. In Bayesian fine-tuning, the user provides a range of values for each hyperparameter, and the console runs the machine learning model for a randomized set of hyperparameters and changes the hyperparameters based on the performance of the model for previous hyperparameters [38]. Since our study deals with a large dataset, we used Bayesian optimization tuning to deal with noise and reduce run times.

**Statistical analysis.** To measure the association of independent variables with asthma development in youth, we selected the variables in the intersection of the two variable lists identified from LASSO and Random Forest. We conducted two types of statistical analysis, namely a Chi-square test for categorical variables and the univariate logistic regression model for numerical variables, to determine if the variables were significantly associated with asthma development in youth. Furthermore, we calculated the mean SHAP (Shapley Additive exPlanations) values to determine the contribution of each independent variable to the prediction of asthma development. We accomplished this by identifying variables with the greatest magnitude of its mean SHAP value, indicating that the variable is highly influential in the prediction (whether or not the person has asthma).

**External validation of machine-learning models.** We conducted external validation of our machine-learning models using the most recent 2023 NHIS dataset. We pre-screened and pre-processed the 2023 NHIS dataset, resulting in 4,919 samples, using the same methods and variables as those applied to the 2021 and 2022 datasets. Compared to the 2021 and 2022 data, six variables were missed in the 2023 NHIS data, including "Ever had COVID-19 (CVDDIAG_C)", "Symptoms of COVID-19 (CVDSEV_C)", "Ever lost consciousness (TBILOSTCON_C)", "Ever dazed or memory gap (TBIDAZED_C)", "Ever headache, vomit, blurred vision, or mood change after blow to head (TBIHEADSYM_C)", and "Ever checked for concussion (TBICHKCONC_C)". Using the 2023 dataset, we validated the machine-learning models trained on the 2021 and 2022 under-sampled dataset.

## Ethics statement

The National Center for Health Statistics' Research Ethics Review Board and the U.S. Office of Management and Budget approved the NHIS study. All respondents in the NHIS survey study provided oral consent. The NHIS data are all deidentified and publicly available from the Centers for Disease Control and Prevention (CDC) website and both authors declare no competing interests.

## Results

### Feature selection

After data pre-processing, the 2021 and 2022 combined NHIS dataset contained 9,716 youth subjects with 45 variables. Two different but complementary feature selection models were employed to identify important variables that might be

associated with asthma development in youth: LASSO and Random Forest. This resulted in 9,716 youth subjects with 39 variables. Among 9,716 youth subjects, 1,043 subjects (10.7%) have reported asthma, and 8,673 subjects (89.3%) do not have reported asthma. To avoid multicollinearity, we removed 5 more variables, which resulted in 34 variables utilized in model training. Through Random Forest, 13 variables were identified, such as age, gender, and family poverty ratio, which are also included in the variable list from LASSO (S1 Table). To maximize the predictive ability of machine learning models and minimize the false negatives, we took the union of selected variables from two lists, which resulted in 34 variables in total.

**Predictive models for asthma development in youth**

Table 1 summarizes the performances of the five predictive models on the 2021 and 2022 tested data, including random forest, XGBoost, neural networks, logistic regression, and support vector machine (SVM) with the linear kernel. Trained with the original unbalanced data without any sampling techniques, XGBoost showed the best model performance with AUC and F1 scores of 0.6796 and 0.2022. In contrast, while it has the highest accuracy, SVM (linear) had no precision and sensitivity, especially the F1 score, indicative of poor model performance.

We applied the undersampling technique to address the class imbalance issue (only 10.3% of subjects with asthma) in the dataset, which randomly removed samples from the major class (subjects without asthma) to match the number of subjects as the minor class (subjects with asthma). As shown in Table 1, all machine learning models trained on the balanced dataset with undersampling performed much better than those trained on the original data, as evidenced by the model performance measures, specifically the F1 score (the most appropriate model performance measure for class

Table 1.  Model performance measures of machine learning models for asthma with different sampling techniques (Only the best performances are shown).

| Sampling Technique | Predictive Model | AUC Score | Precision | Sensitivity | Accuracy | F1 Score |
|---|---|---|---|---|---|---|
| No sampling | Random Forest | 0.7460 | 0.4706 | 0.0252 | 0.8906 | 0.0478 |
| | XGBoost | 0.6796 | 0.3358 | 0.1447 | 0.8755 | 0.2022 |
| | Neural Network | 0.5449 | 0.3711 | 0.1132 | 0.8823 | 0.1735 |
| | Logistic Regression | 0.7689 | 0.4082 | 0.0629 | 0.8878 | 0.1090 |
| | SVM (linear) | 0.7503 | 0.0000 | 0.0000 | 0.8909 | 0.0000 |
| Undersampling | Random Forest | 0.7461 | 0.2157 | 0.6384 | 0.7074 | 0.3225 |
| | XGBoost | 0.7573 | 0.2298 | 0.6352 | 0.7280 | 0.3375 |
| | Neural Network | 0.6640 | 0.2602 | 0.5031 | 0.7897 | 0.3430 |
| | Logistic Regression | 0.7654 | 0.2325 | 0.6698 | 0.7228 | 0.3452 |
| | SVM (linear) | 0.7600 | 0.2360 | 0.6509 | 0.7321 | 0.3464 |
| Oversampling | Random Forest | 0.7483 | 0.4032 | 0.0786 | 0.8868 | 0.1316 |
| | XGBoost | 0.6838 | 0.3893 | 0.1604 | 0.8810 | 0.2272 |
| | Neural Network | 0.6146 | 0.2951 | 0.3239 | 0.8419 | 0.3088 |
| | Logistic Regression | 0.6347 | 0.2378 | 0.4434 | 0.7842 | 0.3095 |
| | SVM (linear) | 0.7100 | 0.2687 | 0.3962 | 0.8165 | 0.3202 |
| SMOTE-TOMEK | Random Forest | 0.7492 | 0.4333 | 0.0818 | 0.8882 | 0.1376 |
| | XGBoost | 0.6738 | 0.3261 | 0.1415 | 0.8744 | 0.1974 |
| | Neural Network | 0.5835 | 0.2890 | 0.2390 | 0.8528 | 0.2616 |
| | Logistic Regression | 0.6949 | 0.2331 | 0.4434 | 0.7801 | 0.3055 |
| | SVM (linear) | 0.6996 | 0.2394 | 0.4245 | 0.7901 | 0.3061 |

Note: The heat map colors compare values within each column.

imbalance). For example, while the AUC and F1 scores for the neural networks model trained on original data are 0.5449 and 0.1735, the model performance was significantly increased when the model was trained on the under-sampled dataset, with the AUC and F1 score of 0.6640 and 0.3430. In addition, trained on the balanced dataset, the logistic regression (AUC = 0.7654 and F1 = 0.3452) and support vector machine (AUC = 0.7600 and F1 = 0.3464) performed well based on their AUC and F1 scores.

To deal with the class imbalance, another sampling technique, oversampling, was applied to oversample the minority group (subjects with asthma). Based on the F1 score and AUC value, as shown in Table 1, the oversampling technique improved the model performance compared to the models trained on the original dataset. For example, the AUC and F1 scores for the neural networks model are 0.6146 and 0.3088. However, the performance of the models trained on the oversampled dataset was poorer than that of those trained on the under-sampled dataset. In addition, the combination of both oversampling and undersampling using SMOTE-Tomek technique resulted into similar model performance with that of those trained on the oversampling technique, but lower than that of those trained on the under-sampled dataset (Table 1).

Out of all the sampling techniques and models, the models trained on the undersampling of the major class yielded the best model performance. In addition, logistic regression and support vector machine were the two models with the best model performance based on the AUC and F1 scores.

To ensure the generalizability, reliability, and robustness of our machine-learning models trained on the 2021 and 2022 data, we utilized the NHIS 2023 dataset for external validation of these models. As shown in Table 2, the performance of each model on the 2023 dataset was similar but lower than the performance of each model tested on the 2021 and 2022 under-sampled dataset. For example, the AUC and F1 scores for logistic regression using 2023 NHIS data were 0.7464 and 0.3223, which are comparable to the model performance on the 2021 and 2022 under-sampled data, 0.7654 and 0.3452.

To assess differences in predictive model performance across age groups, we applied the models to three cohorts: 0–4, 5–12, and 13–17 years, using the undersampled dataset. S2 Table demonstrated that, for the 2021 and 2022 datasets, all five models achieved higher performance metrics, such as AUC and F1 scores, in the 5–12 age group compared to the 0–4 and 13–17 groups. S3 Table confirmed these findings with the validation dataset from 2023 BRFSS data. These results indicate that predictive accuracy for asthma is the greatest in the 5–12 age group.

## Potential risk factors for asthma development in youth

While it is important to predict the risk of asthma development in youth using machine learning models, it is equally important to identify factors that might contribute to asthma development, which can be valuable for early prevention. Using Random Forest and LASSO, we identified two lists of variables that might be the potential risk factors for asthma development. To test how these candidate variables are associated with asthma development, we selected the intersection of variables between two lists to avoid possible false positives and tested for their effects on asthma development. As shown in Table 3, as indicated by the significant P-value ($p < .001$), the occurrence of asthma is highly correlated with sex, age,

**Table 2. External validation of machine-learning models for asthma using 2023 NHIS data.**

| Predictive Model | AUC Score | Precision | Sensitivity | Accuracy | F1 Score |
|---|---|---|---|---|---|
| Random Forest | 0.7187 | 0.1958 | 0.6420 | 0.6633 | 0.3001 |
| XGBoost | 0.7430 | 0.2164 | 0.6401 | 0.6989 | 0.3234 |
| Neural Network | 0.6446 | 0.2371 | 0.4882 | 0.7658 | 0.3191 |
| Logistic Regression | 0.7464 | 0.2114 | 0.6781 | 0.6794 | 0.3223 |
| SVM (linear) | 0.7480 | 0.2170 | 0.6510 | 0.6967 | 0.3255 |

Note: The heat map colors compare values within each column.

the child's health status, number of times the child visited urgent care in the past twelve months, number of times the child visited the emergency room in the past twelve months, if the child took prescription medication in the past twelve months, received a flu vaccine in the past twelve months, the asthma status of the child's parents, the poverty level of the child's family, and the symptoms of COVID-19. Based on the data from all subjects shown in Table 3, males are more likely to develop asthma, children around age 11 are more likely to develop asthma than children around age 9, children with fair to poor health are more prone to asthma, children who visited the emergency room or urgent care more than 3 times are likely to have asthma, children whose parents had asthma are more likely to have asthma.

Fig 1 is the SHAP plot of the most influential features on the prediction of the logistic regression, which depicts how different variations within each feature impacted the model prediction. In this study, no asthma was represented by 0 and 2 and asthma was represented by 1. Therefore, features with a large positive SHAP value pushed the prediction towards asthma while features with a large negative SHAP value pushed the prediction towards no asthma. As shown in Fig 1, took prescription medication in past 12 months, age, and general health status are the three most influential features in asthma risk prediction. In addition, we utilized the mean SHAP values for all selected variables to examine the contribution of each variable to the asthma prediction. As shown in S1 Table, the variables "took prescription medication in past 12 months", "age" and "general health status" had the highest magnitude mean SHAP value of 0.094132, 0.076407 and 0.042564. Therefore, these variables are the most important variables in our models contributing to the prediction of asthma.

## Discussion

This study utilizes the data from the 2021 and 2022 youth and parent National Health Interview Survey to build machine-learning models to predict asthma development in youth. The NHIS includes questions ranging from pediatric to demographic and socioeconomic information. We linked the youth survey data with their parent survey data to account for the contribution of parents' information to asthma development in youth. We applied several sampling techniques to deal with the class imbalance issue (most subjects without asthma), including oversampling, undersampling, and both. We built five different machine learning models to predict asthma development in youth. By comparison, predictive models trained on the dataset with the undersampling of the majority class yielded the best model performance overall. Undersampling outperformed no sampling and oversampling by balancing the training dataset without causing overfitting or creating synthetic data points that may not represent real-world scenarios. Among the models tested, the logistic regression and support vector machine with a linear kernel were the best-performing models for predicting asthma development in youth, based on the AUC score and especially the F1 score. This may be due to the relatively small size of the NHIS dataset after pre-processing and the goal of this study being binary classification, in which these two models can perform well. In this study, the relatively poor model performance of neural network and XGBoost were due to underfitting. In contrast, the poor model performance of random forest largely resulted from overfitting. Therefore, in this study, with the NHIS dataset, the simple models, logistic regression and SVM (linear), might be more suitable for predicting the risk of asthma in youth than other more complex machine learning models. In addition, we found that several factors are significantly associated with asthma development in youth, such as age, gender, and family poverty ratio.

In this study, by employing different sampling techniques to address the class imbalance issue, we have successfully built multiple machine learning models to predict asthma development in youth aged 0–17 with good model performance. One previous study used machine learning to predict asthma amongst children aged 7 months to 12 years at a Morocco hospital, which achieved high performances with F1 scores around 0.80 [9]. However, prenatal, perinatal, postnatal, and environmental factors cannot directly apply to the US population and children aged 13–17, which might limit its generalization. A recent study used Canadian Healthy Infant Longitudinal Development birth cohort data to predict pediatric asthma [29]. The dataset was focused on children up to 4 years of age and included family medical history, clinical data, and environmental factors for young children. They used machine-learning models with 1,484 children and 132 variables

**Table 3. Summary statistics on the variables identified by both LASSO and random forest.**

| Variable | Asthma (n = 1043) | No Asthma (n = 8673) | Chi-square | *P*-value |
|---|---|---|---|---|
| **Sex** | | | 33 | <.001 |
| Male | 616 (12.5%) | 4301 (87.5%) | | |
| Female | 427 (8.9%) | 4372 (91.1%) | | |
| **Age (mean ± SD)** | 11.3 ± 4.1 | 9.5 ± 4.7 | | <.001* |
| **General Health Status** | | | 420.9 | <.001 |
| Excellent | 416 (6.6%) | 5880 (93.4%) | | |
| Very Good | 327 (14.8%) | 1882 (85.2%) | | |
| Good | 233 (23%) | 780 (77%) | | |
| Fair | 60 (34.9%) | 112 (65.1%) | | |
| Poor | 7 (26.9%) | 19 (73.1%) | | |
| **Number of times visited urgent care in past 12 months** | | | 36.2 | <.001 |
| 0 times | 711 (9.8%) | 6538 (90.2%) | | |
| 1 time | 172 (12.3%) | 1230 (87.7%) | | |
| 2 times | 91 (13.7%) | 575 (86.3%) | | |
| 3 times | 30 (15.2%) | 167 (84.8%) | | |
| 4 times | 21 (17.8%) | 97 (82.2%) | | |
| 5+ times | 18 (21.4%) | 66 (78.6%) | | |
| **Number of times visited emergency room in past 12 months** | | | 79.9 | <.001 |
| 0 times | 824 (9.8%) | 7612 (90.2%) | | |
| 1 time | 144 (15.8%) | 770 (84.2%) | | |
| 2 times | 48 (17.4%) | 228 (82.6%) | | |
| 3 times | 14 (30.4%) | 32 (69.6%) | | |
| 4+ times | 13 (29.5%) | 31 (70.5%) | | |
| **Took prescription medicine in the past 12 months** | | | 440.2 | <.001 |
| Yes | 635 (20.3%) | 2490 (79.7%) | | |
| No | 408 (6.2%) | 6183 (93.8%) | | |
| **Received flu vaccine in past 12 months** | | | 7.05 | .008 |
| Yes | 538 (11.6%) | 4092 (88.4%) | | |
| No | 505 (9.9%) | 4581 (90.1%) | | |
| **Parent ever had asthma** | | | 142.2 | <.001 |
| Yes | 272 (20.1%) | 1083 (79.9%) | | |
| No | 771 (9.2%) | 7590 (90.8%) | | |
| **Family poverty ratio (mean ± SD)** | 3.5 ± 2.7 | 3.9 ± 2.9 | | <.001* |
| **Symptoms of COVID-19** | | | 31.5 | <.001 |
| No COVID-19 | 777 (10.2%) | 6877 (89.8%) | | |
| No Symptoms with COVID-19 | 50 (12.8%) | 342 (87.2%) | | |
| Mild Symptoms with COVID-19 | 134 (11.2%) | 1061 (88.8%) | | |
| Moderate Symptoms with COVID-19 | 63 (15.8%) | 336 (84.2%) | | |
| Severe Symptoms with COVID-19 | 19 (25.0%) | 57 (75.0%) | | |
| **Parent ever smoked a cigar** | | | 0.02 | 0.88 |
| Yes | 293 (10.6%) | 2461 (89.4%) | | |
| No | 750 (10.8%) | 6212 (89.2%) | | |
| **Urban-rural classification** | | | 1.67 | 0.64 |
| Large central metro | 299 (10.2%) | 2646 (89.8%) | | |
| Large fringe metro | 277 (11.1%) | 2216 (88.9%) | | |

*(Continued)*

**Table 3.** (Continued)

| Variable | Asthma | No Asthma | Chi-square | *P*-value |
|---|---|---|---|---|
| | (n = 1043) | (n = 8673) | | |
| Medium and small metro | 320 (10.8%) | 2640 (89.2%) | | |
| Nonmetropolitan | 147 (11.2%) | 1171 (88.8%) | | |
| **Household region** | | | 2.48 | 0.48 |
| Northeast | 151 (10.3%) | 1315 (89.7%) | | |
| Midwest | 204 (10%) | 1832 (90%) | | |
| South | 408 (11.3%) | 3211 (88.7%) | | |
| West | 280 (10.8%) | 2315 (89.2%) | | |

Note: * indicates the P-value from univariate logistic regression model.

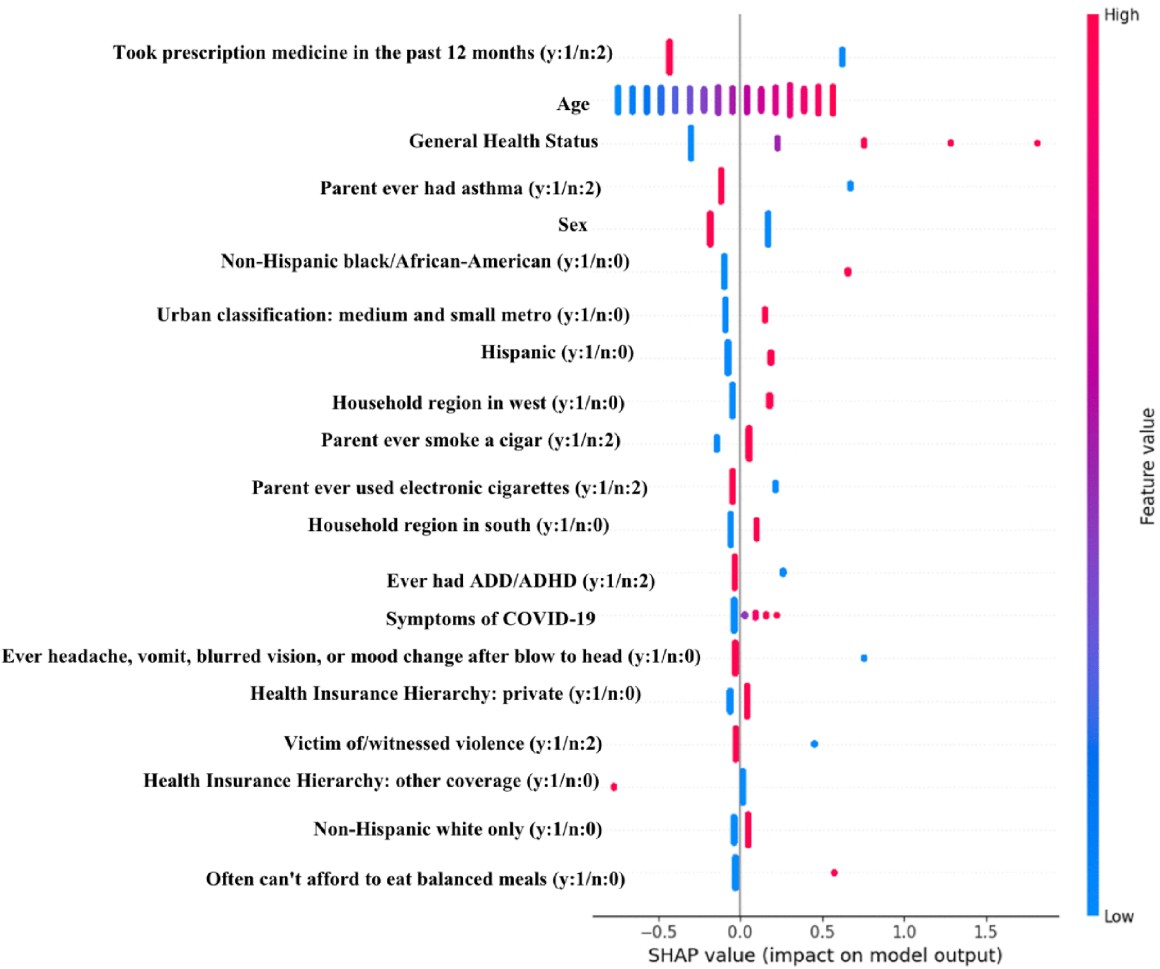

**Fig 1. SHAP summary plot of top 20 variables in the asthma risk prediction model.**

and achieved great model performance with an AUC of 0.99 when predicting asthma in children at age 4 [29]. However, their model cannot apply to children aged 5–17. Further, their data heavily relied on pediatric and prenatal clinical data extracted from patient health records and some environmental factors, which are difficult to obtain in real life, especially for those with low socioeconomic status. The dataset used in our study is national survey data containing basic demographics on youth aged 0–17, which is easily accessible, making it more practical. In this study, we examined model performance across age groups. The predictive models for asthma performed better for ages 5–12 compared to ages 0–4 and 13–17, which might be due to the relatively larger sample size in the 5–12 age group or the data from the 5–12 age group predicting asthma risk more effectively. The exact reason for this finding is still unknown, which requires further investigation.

In another asthma study conducted on the 2019 Michigan Behavioral Risk Factor Surveillance System (BRFSS) data, the group employed similar pre-processing techniques and used SMOTE and ROSE to deal with the class imbalance in the data [8]. That study also employed similar machine-learning models on their dataset, yielding decent performance and identifying many risk factors. However, their dataset focuses on adults in Michigan rather than youth in the United States. Furthermore, the performance of the models in the Michigan study had the highest AUC score of 0.629 and F1 score of 0.287. In our study, the best model has an AUC score of 0.7632 and an F1 score of 0.3416. Therefore, the predictive models in our study performed much better in predicting asthma development in youth, which might be due to the improved pre-processing techniques and a more encompassing data source in this study. Some of the significant variables identified in the Michigan study were also identified in our study, such as flu vaccine and income. While the Michigan study showed that females have a higher risk for asthma development in adults, our study showed that males have a higher risk for asthma development in youth. This disparity is likely due to the different datasets utilized between this study and the Michigan study. The Michigan study used the Michigan BRFSS data, which only includes participants from Michigan who were 18 years or older. In contrast, our study utilizes a national survey mainly focused on participants younger than 18 years of age.

When conducting external validation of the models using the 2023 NHIS data, we noticed a slight decrease in the model performance compared to the 2021 and 2022 data. One potential reason is that six variables included in the trained models were missing in the 2023 NHIS data, which likely contributed to the slightly lower performance of the models using the 2023 NHIS data. These variables such as "Symptoms of COVID-19" and "Ever dazed or memory gap", which were excluded in the 2023 dataset, had a relatively high SHAP value, indicating its influence in the model prediction. Nevertheless, similar performance of models (trained on the 2021 and 2022 under-sampled data) on the 2023 NHIS data further validated the robustness and generalization of our machine learning models in predicting the risk of asthma in youth.

Besides building predictive models for asthma development in youth, another aim of this study is to identify potential risk factors for asthma development in youth, which might help with early asthma detection and prevention. Our study has identified several well-known risk factors for asthma development in youth, such as the presence of asthma in the parent, gender, socioeconomic status, and maternal smoking [39–42]. It is well-known that family history is one of the major causes of asthma [5,39]. This could be due to the shared environment between the parents and child that may influence the development of asthma [43]. Our study also found that boys are at a higher risk for asthma development than girls, which is also consistent with previous studies [40,41]. The increased risk for asthma development in boys might be due to the increased allergic inflammation and serum IgE levels in boys, as well as the smaller airway diameter relative to lung volumes in boys than girls [40,41]. Children living in families with lower socioeconomic status have a higher risk of asthma [42]. A previous review has shown that low socioeconomic status could be a risk factor for pediatric asthma as children living in families with low socioeconomic status have greater exposure to indoor and outdoor hazards (such as molds, mice, second-hand smoke, chemicals, and air pollutants) associated with increased risk of asthma [44]. Additionally, maternal smoking was another factor identified that could influence asthma development in both the mother and, to a greater degree, the child [43]. A review study indicated the influence of maternal smoking on childhood asthma that leads

to increased asthma risk in children through direct damage on fetal lung development, fetal growth, and neuronal differentiation and later on modulated airway hyperreactivity and reduced lung function [45].

Based on the SHAP plot and values, the most influential variables in predicting asthma risk among youth include those who took prescription medication in the past 12 months, age, and general health status. It is reasonable to observe the significant association between high asthma risk with taking a prescription medication and poor health status. Asthma patients might be taking medication to help with their asthma and have bad general health status, or the individuals who took medication due to bad health conditions might have a higher risk of asthma. We speculate that those who are older among youth might be more likely to be exposed to environmental risks, which leads to a higher risk of asthma. These variables and other identified variables in this study need to be further investigated and confirmed in future studies.

An interesting finding was that more people with asthma took a flu shot in the past 12 months than those who did not. As the Michigan BRFSS study mentioned, this could be due to children with asthma being more prone to receiving the flu shot, as their parents may be aware of the dangers that arise if a child with asthma gets the flu [8]. Another interesting finding is the association of severe COVID-19 symptoms with asthma in youth. Currently, there are mixed results on the association of COVID-19 with asthma. A recent nationwide population-based cohort study with approximately 50 million people in South Korea showed that COVID-19 was associated with an increased risk of asthma onset [46]. A retrospective cohort study on 27,423 children in Children's Hospital of Philadelphia Care Network ages 1–16 from March 1, 2020, to February 28, 2021, showed that the COVID-19 diagnosis has no significant impact on asthma onset [47]. However, that study did not examine whether severe symptoms of COVID-19 could impact asthma onset in children. Using national survey data, our study is the first to provide the association between severe COVID-19 symptoms and asthma risks in U.S. youth. Our observation that severe COVID-19 symptoms were associated with a higher risk of asthma (25.0% vs. 10.2%) underscores the potential long-term respiratory impacts of viral illness in children. These findings align with emerging evidence of post-viral airway remodeling [48], which deserves further investigation and highlights the importance of ongoing surveillance, preventive strategies, and public health interventions to mitigate pediatric asthma burden. In addition, we observed a modest decline in model performance when the COVID-19 variable was absent from the external validation dataset (NHIS 2023), highlighting its important contribution to asthma prediction in the pediatric population. Evidence from a recent multicenter retrospective cohort study of 274,803 adults in the U.S. Collaborative Network of the TriNetX database, a large real-world clinical dataset, demonstrated a strong association between COVID-19 infection and subsequent asthma onset [49]. Similar validation efforts could be extended to pediatric populations by leveraging real-world clinical health record data to further evaluate the relationship between COVID-19 and asthma onset in children.

Based on easily accessible demographic and family history data, our predictive models have the potential to be implemented in an easily accessible online asthma risk calculator, which would allow any users to obtain a pre-screening before seeking professional medical diagnosis if at high risk of asthma. Furthermore, individuals at high risk of asthma as predicted by our models could be directed to a wearable respiratory monitoring device, which could more accurately diagnose or evaluate the risk of asthma for that individual [50–52]. Incorporating easily accessible, parent-reported variables—such as overall health status, urgent care visits, flu vaccination history, parental asthma history, and prescription medication use—into routine screening could strengthen early identification of children at risk for asthma in real-world settings. Leveraging these low-burden measures can enable large-scale, cost-effective screening approaches and guide resource allocation in high-volume primary care environments.

This study has several limitations. First, the NHIS does not contain every factor potentially associated with asthma, as its purpose is to get a general health overview of the public. This prevents us from identifying more significant variables associated with asthma development in youth. Including those variables might lead to better model performance. Another limitation is the lack of detail in the prescription medication question. The NHIS survey only asked whether the selected child had taken any prescription medication in the past 12 months, without specifying the type. Consequently, we cannot determine whether asthma medications were included among the reported prescriptions and are unable to

exclude children taking asthma medications from our machine learning models. Second, undersampling could have removed many records, which might negatively influence the model performance. Third, although the predictive models have relatively better model performance than previous studies, the models need to be further improved, for example, by including more variables. Fourth, due to the cross-sectional nature of the dataset in this study, we are unable to determine the causal effect, the potential risk factors for asthma development in youth identified in this study must be further validated by longitudinal studies. Fifth, our use of LASSO in feature selection may not have been the best option as LASSO is a regression tool. In contrast, the logistic regression model with an L1 penalty is a classification tool, which may be more beneficial for identifying factors associated with asthma. Sixth, a small percentage of overlaps might exist between the 2021 and 2022 samples. However, the NIHS is a cross-sectional national survey study, and the study redesign after 2019 allows data users to pool multiple years of sample together to obtain reliable estimates [53]. Thus, we expect the potential overlaps won't affect our analysis results. Lastly, employing sampling techniques in our study may lead to less calibrated models, which would undermine the interpretation of model predictions. When extra data becomes available in the future, we will perform the model calibration. The findings from this study need to be validated using an external dataset with similar variables if such a dataset becomes available in the future. More importantly, our results might be subject to self-reported bias due to the recall error in the survey study. For example, parent-reported asthma diagnoses were not validated by spirometry, and prior evidence suggests that approximately 30% of asthma diagnoses may be inaccurate, often reflecting conditions such as viral wheeze or gastroesophageal reflux disease (GERD), which could lead to an overestimation of pediatric asthma prevalence [54]. Future studies incorporating spirometry validation would strengthen diagnostic accuracy and reduce misclassification bias. Additionally, the absence of COVID-19–related variables in our external validation dataset may have introduced further bias into our findings.

## Conclusions

Using the combined national survey data and applying different sampling techniques, this study successfully built several machine-learning models for predicting asthma development in youth, which will be very valuable for early screening. Out of these models, the logistic regression model and Support Vector Machine with a linear kernel combined with undersampling showed the best model performance. Furthermore, external validation of our models using the NHIS 2023 dataset shows that our models are generalizable, reliable, and robust. The identification of potential risk factors associated with asthma development in youth (such as general health status and having received flu vaccine in the past 12 months) will provide valuable insights for early asthma prevention.

## Supporting information

**S1 Fig.  2021 and 2022 NHIS combined dataset preparation with number of subjects in each step prior to data curation.**
(TIF)

**S2 Fig.  Data pre-processing and feature selection for variables highly associated with asthma in youth.**
(TIF)

**S3 Fig.  Building machine learning models and incorporating sampling techniques for asthma prediction in youth.**
(TIF)

**S1 Table.  Feature selection results by LASSO and Random Forest with absolute value of mean SHAP values calculated through the logistic regression model with undersampling.**
(DOCX)

**S2 Table. Model performance measures of machine learning models for asthma using the 2021 and 2022 NHIS undersampled data for different age groups.**
(DOCX)

**S3 Table. External validation of machine learning models for asthma using the 2023 NHIS undersampled data for different age groups.**
(DOCX)

## Author contributions

**Conceptualization:** Matthew Xie.

**Data curation:** Matthew Xie.

**Formal analysis:** Matthew Xie.

**Investigation:** Matthew Xie.

**Methodology:** Matthew Xie.

**Project administration:** Matthew Xie.

**Supervision:** Chenliang Xu.

**Validation:** Matthew Xie.

**Visualization:** Matthew Xie.

**Writing – original draft:** Matthew Xie.

**Writing – review & editing:** Matthew Xie, Chenliang Xu.

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
