## [Decision Letter · Decision Letter 0]

17 Aug 2025

Dear Dr. Xu,

Thank you for submitting your manuscript to PLOS ONE. After careful consideration, we feel that it has merit but does not fully meet PLOS ONE’s publication criteria as it currently stands. Therefore, we invite you to submit a revised version of the manuscript that addresses the points raised during the review process.

We look forward to receiving your revised manuscript.

Kind regards,

Marsa Gholamzadeh, PhD

Academic Editor

PLOS ONE

Journal Requirements:

Reviewer's Responses to Questions

**Comments to the Author**

1. Is the manuscript technically sound, and do the data support the conclusions?

Reviewer #1: Partly

Reviewer #2: Yes

2. Has the statistical analysis been performed appropriately and rigorously?

Reviewer #1: N/A

Reviewer #2: I Don't Know

3. Have the authors made all data underlying the findings in their manuscript fully available?

Reviewer #1: Yes

Reviewer #2: Yes

4. Is the manuscript presented in an intelligible fashion and written in standard English?

Reviewer #1: Yes

Reviewer #2: Yes

Reviewer #1: I congratulate the authors of this article for conducting this research. I hope that professors and researchers will review the articles with a positive perspective so that the results of a research project that advances science and public awareness will be easily accessible. Below, I have suggested some tips to writers that I hope will be helpful.

1. Use more up-to-date information.

2. A comprehensive review should be done on referencing, as many of the references are not reported.

3. Use newer articles.

4. Make the introduction more concise and also report the results more simply.

5. A heat map is also suggested to enhance this article further.

Reviewer #2: Manuscript ID: PONE-D-25-09363

Predicting Risk of Asthma in Young People through Machine Learning Models

Dear Dr. Xu and Mr. Xie,

I appreciate the chance to offer feedback on asthma risk prediction. I'm thankful, as a clinician, to note your focus on early identification according to national surveys, as undiagnosed asthma has such a strong impact on lung development and quality of life. Your use of unusual predictors (e.g., urgent care visits, prescription rates) is intriguing, and the pragmatic application of undersampling to address class imbalance is great.

Strengths

Your model's application of easily accessible variables (e.g., parent-provided health, flu vaccination history) strengthens real-world screening. These "red flags" would target high-risk children for spirometry or specialist consultation in high-volume primary care clinics. The observed correlation between bad COVID-19 disease and asthma risk (25% prevalence versus 10% in controls) aligns with new findings of post-viral airway remodeling in children, a finding deserving further study.

Clinical Issues & Recommendations

Prescription Drug Variability

Issue: The best predictor—" took prescription medication in the past 12 months" (SHAP 0.094)—is susceptible to circularity. If this includes asthma drugs (e.g., albuterol/ICS), the model could "predict" asthma in already diagnosed/treated children.

Recommendation: Exclude asthma-specific medications or stratify prescriptions according to drug class (e.g., respiratory vs. antibiotics). Define whether medication use reflects comorbidities (e.g., recurrent bronchitis) as opposed to early treatment of asthma.

Age Variability

Problem: Pooling all ages (0–17 years) conceals developmentally different asthma phenotypes (e.g., toddler viral-induced wheeze vs. adolescent obesity/work exposures).

Solution: Use strata analyses by age groups (e.g., <5, 5–12, >12 years) or add phenotype markers (e.g., "eczema diagnosis," "allergy medications") as surrogates.

Clinical Translation & Model Performance

Problem: Good AUC (0.76) but low PPV (0.23) would result in 77% of "high-risk" alarms being false positives and draining resources. 33% of true cases would be missed by sensitivity (0.67).

Solution:

Apply the model as a tier-1 screener, not a diagnosis.

Add actionable pathways (e.g., "High-risk youth should have symptom questionnaires + peak flow monitoring pre-specialist referral").

Add the negative predictive value (NPV)—presumably high with asthma prevalence—to reassure low-risk families.

Data Bias

Problem: Parent-reported "doctor-diagnosed asthma" is not validated (e.g., spirometry reversibility). ~30% of such diagnoses are mistaken (e.g., confusion with viral wheeze/GERD), overestimating false positives. Recall bias (e.g., "ER visits") makes findings difficult.

Solution: Highlight this weakness prominently and validate against objective outcomes datasets (e.g., spirometry) in follow-up studies.

Bridging Algorithmic Promise to Clinical Impact

Refine medication variables to differentiate pre-asthma signals from full-blown disease.

Investigate COVID-19 interactions: Could severe infections reveal genetic susceptibilities?

Enhance interpretability: Add a table converting SHAP values into prevention interventions (e.g., "Fair/poor health children + parental asthma should receive flu vaccination/trigger avoidance counseling").

Revision Recommendations

Significant Revisions:

Exclude asthma medications from prescription variables and re-fit models.

Stratify analyses by age subgroups (0–4, 5–12, 13–17 years).

Reorganize clinical utility: Emphasize screening role, report NPV, and suggest cost-benefit thresholds.

Account for the loss of COVID-19 variables in external validation and sketch clinical validation steps.

Minor Revisions

Discussion: Explain why less complex models (logistic regression/SVM) worked better than the more complicated ones (NN/XGBoost).

Limitations: Discuss self-report bias (e.g., no spirometry validation) and missing COVID-19 variables in external validation.

These clinically informed improvements would allow pediatricians to act sooner and more precisely. I recommend substantial revisions to strengthen the model regarding validity and translational utility.

**Do you want your identity to be public for this peer review?** For information about this choice, including consent withdrawal, please see our Privacy Policy

Reviewer #1: **Yes: ** Dr. Vahid Kazemizadeh

Reviewer #2: **Yes: ** Niloofar Khoshnam Rad

---

## [Author Response · Author response to Decision Letter 1]

23 Sep 2025

PONE-D-25-09363

Predicting the Risk of Asthma Development in Youth Using Machine Learning Models

PLOS ONE

We appreciate the reviewers’ many suggestions and comments. We have revised our manuscript to incorporate the reviewers’ comments and suggestions. The details are listed below.

>> Reviewer #1: I congratulate the authors of this article for conducting this research. I hope that professors and researchers will review the articles with a positive perspective so that the results of a research project that advances science and public awareness will be easily accessible. Below, I have suggested some tips to writers that I hope will be helpful.

>> 1. Use more up-to-date information.

>> 2. A comprehensive review should be done on referencing, as many of the references are not reported.

>> 3. Use newer articles.

Responses: Thanks so much for the insightful comments! As suggested, we have provided more up-to-date information and cited more recent articles in our revised manuscript.

>> 4. Make the introduction more concise and also report the results more simply.

Responses: Thanks so much for the comments! As suggested, we have revised the introduction and results sections to make them more concise.

>> 5. A heat map is also suggested to enhance this article further.

Responses: Great suggestions! We have incorporated the heatmap into Tables 1 and 2 to enhance their visualization.

>> Reviewer #2: Manuscript ID: PONE-D-25-09363

>> Predicting Risk of Asthma in Young People through Machine Learning Models

>> Dear Dr. Xu and Mr. Xie,

>> I appreciate the chance to offer feedback on asthma risk prediction. I'm thankful, as a clinician, to note your focus on early identification according to national surveys, as undiagnosed asthma has such a strong impact on lung development and quality of life. Your use of unusual predictors (e.g., urgent care visits, prescription rates) is intriguing, and the pragmatic application of undersampling to address class imbalance is great.

>> Strengths

>> Your model's application of easily accessible variables (e.g., parent-provided health, flu vaccination history) strengthens real-world screening. These "red flags" would target high-risk children for spirometry or specialist consultation in high-volume primary care clinics. The observed correlation between bad COVID-19 disease and asthma risk (25% prevalence versus 10% in controls) aligns with new findings of post-viral airway remodeling in children, a finding deserving further study.

>> Clinical Issues & Recommendations

>> Prescription Drug Variability

>> Issue: The best predictor—" took prescription medication in the past 12 months" (SHAP 0.094)—is susceptible to circularity. If this includes asthma drugs (e.g., albuterol/ICS), the model could "predict" asthma in already diagnosed/treated children.

>> Recommendation: Exclude asthma-specific medications or stratify prescriptions according to drug class (e.g., respiratory vs. antibiotics). Define whether medication use reflects comorbidities (e.g., recurrent bronchitis) as opposed to early treatment of asthma.

>> Age Variability

>> Problem: Pooling all ages (0–17 years) conceals developmentally different asthma phenotypes (e.g., toddler viral-induced wheeze vs. adolescent obesity/work exposures).

>> Solution: Use strata analyses by age groups (e.g., <5, 5–12, >12 years) or add phenotype markers (e.g., "eczema diagnosis," "allergy medications") as surrogates.

>> Clinical Translation & Model Performance

>> Problem: Good AUC (0.76) but low PPV (0.23) would result in 77% of "high-risk" alarms being false positives and draining resources. 33% of true cases would be missed by sensitivity (0.67).

>> Solution:

>> Apply the model as a tier-1 screener, not a diagnosis.

>> Add actionable pathways (e.g., "High-risk youth should have symptom questionnaires + peak flow monitoring pre-specialist referral").

>> Add the negative predictive value (NPV)—presumably high with asthma prevalence—to reassure low-risk families.

>> Data Bias

>> Problem: Parent-reported "doctor-diagnosed asthma" is not validated (e.g., spirometry reversibility). ~30% of such diagnoses are mistaken (e.g., confusion with viral wheeze/GERD), overestimating false positives. Recall bias (e.g., "ER visits") makes findings difficult.

>> Solution: Highlight this weakness prominently and validate against objective outcomes datasets (e.g., spirometry) in follow-up studies.

>> Bridging Algorithmic Promise to Clinical Impact

>> Refine medication variables to differentiate pre-asthma signals from full-blown disease.

>> Investigate COVID-19 interactions: Could severe infections reveal genetic susceptibilities?

>> Enhance interpretability: Add a table converting SHAP values into prevention interventions (e.g., "Fair/poor health children + parental asthma should receive flu vaccination/trigger avoidance counseling").

Responses: Thanks so much for the insightful comments! We have revised our manuscript according to the reviewers’ comments and suggestions.

>> Revision Recommendations

>> Significant Revisions:

>> Exclude asthma medications from prescription variables and re-fit models.

Responses: Thanks so much for the insightful comments! Unfortunately, there is no variable for asthma medication in the National Health Interview Survey (NHIS) that allows us to exclude asthma medications from the data, as the survey question is “At any time in the past 12 months, did the selected child take prescription medication?”, which includes all the prescribed medications. We have added this as a limitation in our revised manuscript.

“Another limitation is the lack of detail in the prescription medication question. The NHIS survey only asked whether the selected child had taken any prescription medication in the past 12 months, without specifying the type. Consequently, we cannot determine whether asthma medications were included among the reported prescriptions and are unable to exclude children taking asthma medications from our machine learning models.”

>> Stratify analyses by age subgroups (0–4, 5–12, 13–17 years).

Responses: Thanks so much for the insightful comments! We have stratified the analysis by age groups and reported the results in the supplemental file of our revised manuscript (see Supplemental Tables 2 and 3).

Methods section: “In addition, in order to compare the model performance for different age groups, we stratified our data into three age groups: 0-4, 5-12, and 13-17 years.”

Results section: “To assess differences in predictive model performance across age groups, we applied the models to three cohorts: 0-4, 5-12, and 13-17 years, using the under-sampled dataset. Table S2 demonstrated that, for the 2021 and 2022 datasets, all five models achieved higher performance metrics, such as AUC and F1 scores, in the 5-12 age group compared to the 0-4 and 13-17 groups. Table S3 confirmed these findings with the validation dataset from 2023 BRFSS data. These results indicate that predictive accuracy for asthma is the greatest in the 5-12 age group.”

Discussion section: “In this study, we examined model performance across age groups. The predictive models for asthma performed better for ages 5-12 compared to ages 0-4 and 13-17, which might be due to the relatively larger sample size in the 5-12 age group or the data from the 5-12 age group predicting asthma risk more effectively. The exact reason for this finding is still unknown, which requires further investigation.”

>> Reorganize clinical utility: Emphasize screening role, report NPV, and suggest cost-benefit thresholds.

Responses: Thanks so much for the insightful comments! We have added the clinical utility of our findings to the Discussion section of the revised manuscript. We chose not to report the negative predictive value (NPV) because it is highly dependent on asthma prevalence in the study population, which limits its generalizability across different clinical settings. Instead, we focused on prevalence-independent metrics such as AUC, precision, sensitivity, accuracy, and F1 score, which more reliably reflect the performance of our machine learning models. Additionally, our external validation dataset does not include the COVID-19 variable; reporting NPV under these conditions could introduce additional noise and reduce the interpretability of our results.

“Incorporating easily accessible, parent-reported variables—such as overall health status, urgent care visits, flu vaccination history, parental asthma history, and prescription medication use—into routine screening could strengthen early identification of children at risk for asthma in real-world settings. Leveraging these low-burden measures can enable large-scale, cost-effective screening approaches and guide resource allocation in high-volume primary care environments.”

“Our observation that severe COVID-19 symptoms were associated with a higher risk of asthma (25.0% vs. 10.2%) underscores the potential long-term respiratory impacts of viral illness in children. These findings align with emerging evidence of post-viral airway remodeling, which deserves further investigation and highlights the importance of ongoing surveillance, preventive strategies, and public health interventions to mitigate pediatric asthma burden.”

>> Account for the loss of COVID-19 variables in external validation and sketch clinical validation steps.

Responses: Thanks so much for the insightful comments! We have added it to the Discussion section of our revised manuscript as follows:

“We observed a modest decline in model performance when the COVID-19 variable was absent from the external validation dataset (NHIS 2023), highlighting its important contribution to asthma prediction in the pediatric population. Evidence from a recent multicenter retrospective cohort study of 274,803 adults in the U.S. Collaborative Network of the TriNetX database, a large real-world clinical dataset, demonstrated a strong association between COVID-19 infection and subsequent asthma onset (Chuang et al., 2024). Similar validation efforts could be extended to pediatric populations by leveraging real-world clinical health record data to further evaluate the relationship between COVID-19 and asthma onset in children.”

>> Minor Revisions

>> Discussion: Explain why less complex models (logistic regression/SVM) worked better than the more complicated ones (NN/XGBoost).

Responses: Thanks so much for the insightful comments! We have included the following explanation in our manuscript.

“Among the models tested, the logistic regression and support vector machine with a linear kernel were the best-performing models for predicting asthma development in youth, based on the AUC score and especially the F1 score. This may be due to the relatively small size of the NHIS dataset after pre-processing and the goal of this study being binary classification, in which these two models can perform well. In this study, the relatively poor model performance of neural network and XGBoost were due to underfitting. In contrast, the poor model performance of random forest largely resulted from overfitting.”

>> Limitations: Discuss self-report bias (e.g., no spirometry validation) and missing COVID-19 variables in external validation.

>> These clinically informed improvements would allow pediatricians to act sooner and more precisely. I recommend substantial revisions to strengthen the model regarding validity and translational utility.

Responses: Thanks so much for the insightful comments! We have added the limitation in the Discussion section of our revised manuscript.

“For example, parent-reported asthma diagnoses were not validated by spirometry, and prior evidence suggests that approximately 30% of asthma diagnoses may be inaccurate, often reflecting conditions such as viral wheeze or gastroesophageal reflux disease (GERD), which could lead to an overestimation of pediatric asthma prevalence (54). Future studies incorporating spirometry validation would strengthen diagnostic accuracy and reduce misclassification bias. Additionally, the absence of COVID-19–related variables in our external validation dataset may have introduced further bias into our findings.”

---

## [Decision Letter · Decision Letter 1]

29 Oct 2025

Predicting the Risk of Asthma Development in Youth Using Machine Learning Models

PONE-D-25-09363R1

Dear Dr. Xu,

We’re pleased to inform you that your manuscript has been judged scientifically suitable for publication and will be formally accepted for publication once it meets all outstanding technical requirements.

Kind regards,

Marsa Gholamzadeh, PhD

Academic Editor

PLOS ONE
---

## [Editor Report · Acceptance letter]

PONE-D-25-09363R1

PLOS ONE

Dear Dr. Xu,

I'm pleased to inform you that your manuscript has been deemed suitable for publication in PLOS ONE. Congratulations! Your manuscript is now being handed over to our production team.

Kind regards,

on behalf of

Dr. Marsa Gholamzadeh

Academic Editor

PLOS ONE